# Up-regulation of cytosolic prostaglandin E synthase in fetal-membrane and amniotic prostaglandin $E_2$ accumulation in labor

Nanase Takahashi[1,2], Toshiaki Okuno[1]*, Hiroki Fujii[1], Shintaro Makino[3], Masaya Takahashi[3], Mai Ohba[1], Kazuko Saeki[1], Atsuo Itakura[2], Satoru Takeda[2], Takehiko Yokomizo[1]

1 Department of Biochemistry, Juntendo University School of Medicine, Tokyo, Japan, 2 Department of Obstetrics & Gynecology, Juntendo University School of Medicine, Tokyo, Japan, 3 Department of Obstetrics & Gynecology, Juntendo University Urayasu Hospital, Chiba, Japan

* tokuno@juntendo.ac.jp

**Data Availability Statement:** All relevant data are within the paper.

**Funding:** This work was supported by MEXT/JSPS KAKENHI Grant Numbers 15KK0320, 16K08596,

## Abstract

Prostaglandin $E_2$ ($PGE_2$) is known to have important roles in labor, but the detailed mechanism underlying the spontaneous human labor remains unknown. Here, we examined the involvement of prostaglandin biosynthetic enzymes and transporter in the accumulation of $PGE_2$ in amniotic fluid in human labor. $PGE_2$ and its metabolites were abundant in amniotic fluid in deliveries at term in labor (TLB), but not at term not in labor (TNL). In fetal-membrane Transwell assays, levels of $PGE_2$ production in both maternal and fetal compartments were significantly higher in the TLB group than the TNL group. In fetal-membrane, the mRNA level of *PTGES3*, which encodes cytosolic prostaglandin E synthase (cPGES), was significantly higher in TLB than in TNL, but the mRNA levels of the other $PGE_2$-synthase genes were not affected by labor. Moreover, the mRNA level of *PTGS2*, which encodes cyclooxygenase-2 (COX-2) in the amnion was significantly higher in TLB than in TNL. Western blot analyses revealed that the levels of COX-1 and COX-2 were comparable between the two groups, however, the level of cPGES was relatively higher in TLB than in TNL. COXs, cPGES, and prostaglandin transporter (SLCO2A1) proteins were all expressed in both chorionic trophoblasts and amniotic epithelium. These findings suggest that COXs, cPGES and SLCO2A1 contribute to $PGE_2$ production from fetal-membrane in labor.

## Introduction

Prostaglandin $E_2$ ($PGE_2$) is a bioactive lipid mediator and plays important roles in the induction and maintenance of labor in mammals [1,2]. $PGE_2$ is synthesized from arachidonic acid (AA) sequentially by the actions of cyclooxygenase (COX)-1/COX-2 and three terminal $PGE_2$ synthase (PGES) enzymes: cytosolic PGES (cPGES), microsomal PGES (mPGES)-1, and mPGES-2. In addition, $PGE_2$ is an organic anion, and at physiological pH, the PG transporters, such as solute carrier organic anion transporter family member 2A1 (SLCO2A1), are required for its secretion. Although some studies reported the accumulation of $PGE_2$ in amniotic fluid

19K07357 (to T. O.), 15H05904, 15H04708, 18H02627, 19KK0199 (to T.Y.), 18K06923 (to K. S.) and by grants from AMED CREST (20gm1210006), the Mishima Kaiun Memorial Foundation (to TO), the Ono Medical Research Foundation, the Uehara Memorial Foundation (to TY). The study was supported in part by a Grant-in-Aid (S1311011) from the Foundation of Strategic Research Projects in Private Universities from the MEXT, and a grant from the Institute for Environmental and Gender-Specific Medicine (to TY).

**Competing interests:** The authors have declared that no competing interests exist.

from pregnant women with labor [3], the detailed molecular mechanism underlying $PGE_2$ production with labor remains largely unknown. Up-regulation of COX-2 expression has been observed in fetal-membrane, placental villous tissue, and uterine myometrium prior to the onset of spontaneous labor [4–6]. However, only a few studies have investigated the expression of PGESs in human fetal-membrane from pregnant women with labor. One study reported the increased expression of mPGES-1 in chorion from pregnant women with labor [7], but the other study reported no differences of the expression with labor [8]. The expression of mPGES-1 in fetal-membrane from pregnant women with labor is increased by infection [9]. However, the expression of cPGES with labor remains largely unclear [10].

In this study, we collected the TLB (term in labor) samples from the pregnant women with spontaneous onset of labor and the TNL (term not in labor) samples from the women before the onset of labor, and quantified PGs and their metabolites in the amnion, and examined the expression of enzymes and transporters involved in production and release of PGs. To our knowledge, this is the first study to suggest that cPGES contributes to $PGE_2$ production from fetal-membrane in labor.

## Materials and methods

### Study participants

Pregnant women at term ($n$ = 21) were classified according to the type of delivery: TLB ($n$ = 10) or TNL ($n$ = 11). Onset of labor was defined as the presence of regular uterine contractions with a frequency greater than one every 10 min, with cervical changes including softening and opening. All participants had been free of medication for $\geq$2 weeks prior to delivery, whereas those who fulfilled the diagnostic criteria for chorioamnionitis were excluded. The participants with bacterial vaginosis were also excluded, and there were no participants who had meconium in our study.

Umbilical-cord blood and fetal-membrane were collected immediately after delivery of the placenta from each participant. Umbilical-cord blood (2 ml) was collected in a heparinized tube, immediately after delivery. Fetal-membrane (amnion, chorion, and decidua) were collected with avoidance of placental-attachment sites. Fetal-membrane were collected 5 cm from the placenta and sites of membrane rupture, whereas placental tissue was collected 3 cm from the umbilical-cord attachment site. Amniotic fluid was taken by amniocentesis from the TNL group during cesarean sections, and from the TLB group during delivery. All amniotic-fluid samples were free of blood contamination.

### Materials

Eicosanoid standards and deuterium-labeled eicosanoids were purchased from Cayman Chemical (Ann Arbor, MI, USA). All solvents used were HPLC or LC/MS grade, and were purchased from Wako Pure Chemical (Osaka, Japan) or Thermo Fisher Scientific (Waltham, MA, USA).

### Sample preparation for LC-MS/MS analysis

Blood in heparinized tubes was kept on ice and then centrifuged at $800 \times g$ for 5 min at 4˚C to separate plasma from blood cells. Amniotic fluid was centrifuged at $1,000 \times g$ for 10 min at 4˚C. Samples of plasma and supernatants of amniotic fluid were immediately mixed with two volumes of methanol and stored at −80˚C. Samples of fetal-membrane were immediately frozen in liquid nitrogen and stored at −80˚C prior to homogenization with an Automill (Tokken, Chiba, Japan) and extraction of lipids with 1 ml of methanol at −20˚C. After overnight

incubation, the samples were centrifuged at $5,000 \times g$ for 5 min at 4°C. The supernatants were mixed with nine volumes of water containing 0.1% formic acid, which included deuterium-labeled internal standards (Cayman Chemical). Diluted samples were loaded onto a solid-phase extraction cartridge (Waters, Milford, MA, USA) and washed serially with water containing 0.1% formic acid, 15% methanol containing 0.1% formic acid, and petroleum ether containing 0.1% formic acid. After air drying the cartridge with suction, the eicosanoids were eluted with 200 μl of methanol containing 0.1% formic acid.

## LC-MS/MS analysis

Eicosanoid profiling of amniotic fluid and fetal-membranes was performed according to previously described methods, with some modifications [11–13]. Reversed-phase-LC-MS/MS used a Shimadzu liquid-chromatography system consisting of four LC-20AD pumps, a SIL-20AC autosampler, a CTO-20AC column oven, an FCV-12AH six-port switching valve, and a TSQ Quantum Ultra triple-quadrupole mass spectrometer equipped with an electrospray-ionization ion source (Thermo Fisher Scientific). A 50 μl aliquot of each sample was injected into the trap column (Opti-Guard Mini C18) at a total flow rate of 500 μl/min; 3 min after sample injection, the valve was switched to introduce the trapped sample to the analytical column (Capcell Pak C18 MGS3; Shiseido, Tokyo, Japan). Mobile phase A was water, and phase B was acetonitrile:formic acid (1000:1). The gradient conditions consisted of 37% B from 0 to 7 min, a linear gradient of 37–90% B from 7 min to 19 min, 100% B from 19 min to 21 min, and 37% B from 21 min to 22.5 min. The total flow rate was 120 μl/min, the column temperature was set at 46°C, and the column eluent was introduced directly into a TSQ Quantum Ultra mass spectrometer. All compounds were analyzed in a negative-ion-polarity mode. Eicosanoids were quantified by multiple reaction monitoring of the following transitions: $PGE_2$, m/z 351 $\rightarrow$ 271; 15-keto-$PGE_2$, m/z 349 $\rightarrow$ 235; 13,14-dihydro-15-keto-$PGE_2$, m/z 351 $\rightarrow$ 175; 19-hydroxy-$PGE_2$, m/z 367 $\rightarrow$ 189; $[^2H_4]PGE_2$, m/z 355 $\rightarrow$ 275; and $[^2H_4]$13,14-dihydro-15-keto-$PGE_2$, m/z 355 $\rightarrow$ 239. For accurate quantification, calibration curves were generated for each target eicosanoid using known reference standards and the isotope-labeled internal standard. Automated peak detection, calibration, and calculation were carried out with the Xcalibur 2.2 software package.

**Fetal-membrane Transwell assays.** Fetal-membrane Transwell assays were performed as previously described [14]. Briefly, a $3 \times 3$ cm square of fetal-membrane was placed in a 24 mm Transwell clear insert (Corning, Lindfield, Australia) (see Fig 3A) in 6-well culture plate containing serum-free culture medium (DMEM/Ham's Nutrient Mixture F-12, phenol red-free, supplemented with 15 mM HEPES, pH 7.3 (Sigma-Aldrich, St Louis, MO, USA) and 0.5% fatty acid-free BSA (Sigma-Aldrich)). The maternal compartment on the decidua contained 3 ml medium and the fetal compartment on the amnion contained 2.5 ml medium. Membranes were incubated for 120 min at 37°C in an atmosphere of air containing 5% $CO_2$. Medium (100 μl) was collected from both compartments at 0, 15, 30, and 60 min. Eicosanoids in the medium were analyzed by LC-MS/MS as described above.

## RT-qPCR

Fetal-membranes were harvested and placed into RNAlater Solution (Ambion, Austin, TX, USA) at 4°C for 24 h, then stored at −80°C. Total RNA was extracted with RNeasy Fibrous Tissue Kit (Qiagen, Venlo, the Netherlands), and cDNA was generated with QuantiTect Reverse Transcription Kit (Qiagen). qPCR was performed with ABI TaqMan Fast Advanced Master Mix (Thermo Fisher Scientific) on a 7500 Fast Real-Time PCR System (Thermo Fisher Scientific). Gene expression levels were normalized to those of *ACTB* (β-actin) using the ΔΔCt

method. The following Taqman probes were used: *ACTB* (Assay ID: Hs99999903_m1, Gene ID: Hs.520640), *PLA2G4A* (Assay ID: Hs00996912_m1, Gene ID: Hs497200), *PTGS1* (Assay ID: Hs00377726_m1, Gene ID: Hs201978), *PTGS2* (Assay ID: Hs00153133_m1, Gene ID: Hs196384), *PTGES* (Assay ID: Hs01115610_m1, Gene ID: Hs146688), *PTGES2* (Assay ID: Hs00228159_m1, Gene ID: Hs495219), and *PTGES3* (Assay ID: Hs00832847_gH, Gene ID: Hs50425).

## Western blotting

Each fetal-membrane sample was homogenized with an Automill (Tokken, Chiba, Japan) in 2 ml of lysis buffer (50 mM Tris-HCl, pH 7.4, 1% SDS, 50 mM NaCl, 20 mM NaF, 1 mM Na$_3$VO$_4$, 1 mM EDTA, with protease inhibitors (Nacalai Tesque, Kyoto, Japan)). Lysates were centrifuged at $4,000 \times g$ for 5 min at 4˚C, supernatants were recovered, and their protein concentrations were determined with a BCA assay kit (Nacalai Tesque). Each sample was diluted 9:1 with SDS buffer (0.125 M Tris-HCl, pH 6.5, 5% SDS, 25% glycerol, 7.5% 2-mercaptoethanol), and proteins were denatured by incubation at 95˚C for 5 min. Protein samples were separated on 12% SDS-polyacrylamide gels and transferred to PVDF membranes, which were blocked with Tris-buffered saline (TBS) containing 5% dried skimmed milk and then incubated with primary antibody against COX-1 (sc-1752, 1:1,000 dilution in TBS containing 5% dried skimmed milk; Santa Cruz Biotechnology), COX-2 (sc-1745, 1:1,000 dilution in TBS containing 5% dried skimmed milk; Santa Cruz Biotechnology), and cPGES (ab92503, 1:1,000 dilution in TBS containing 5% dried skimmed milk; Abcam), followed by HRP conjugated secondary antibody (SA00001-4, 1:2000 dilution in TBS containing 5% dried skimmed milk; PROTEINTECH GROUP, IN) for COX-1 and COX-2, and HRP-conjugated secondary antibody (NA934, 1:5,000 dilution in TBS containing 5% dried skimmed milk; GE Healthcare, Little Chalfont, UK) for cPGES. Membranes were washed with TBS-T (0.1% Tween 20 in TBS), and antibody-bound protein was visualized with a chemiluminescence reagent kit (Perkin-Elmer, Branchburg, NJ, USA) and LAS-4000 imaging system (FujiFilm, Tokyo, Japan). The membranes were re-blotted with a primary antibody against β-actin (sc-69879, 1:500 dilution in TBS containing 5% dried skimmed milk; Santa Cruz Biotechnology) and HRP-conjugated secondary antibody (NA931V, 1:5,000 dilution in TBS containing 5% dried skimmed milk; GE Healthcare). Chemiluminescence was measured with LAS 4000 film (FujiFilm).

## Immunohistochemical analysis

Fetal-membranes were collected after delivery and processed for histological analysis. Tissues were fixed in 10% formalin/PBS and embedded in paraffin, then 10 μm-thick sections were cut and fixed on New Silane III glass slides (Muto Pure Chemical, Tokyo, Japan). Immunohistochemical staining was performed with a Benchmark GX system (Ventana, Tucson, AZ, USA). The following primary antibodies were used: mouse anti-COX-1 (sc-19998, 1:30 dilution; Santa Cruz Biotechnology, Dallas, TX, USA); mouse anti-COX-2 (sc-166475, 1:150; Santa Cruz Biotechnology); mouse anti-p23 (ab92503, 1:500; Abcam, Cambridge, MA, USA); and rabbit anti-SLCO2A1 (bs-4710R, 1:250; Bioss Antibodies, Woburn, MA, USA). Secondary antibodies were anti-rabbit IgG (1:3,000; DAKO, Santa Clara, CA, USA) and anti-mouse IgG (1:3,000; DAKO). The OptiView Amplification Kit and OptiView DAB IHC Detection Kit (Ventana) were used for detection. Mouse monoclonal immunoglobulin (Ventana) was used as a negative control. Hematoxylin was used for nuclear counterstaining, and the images were taken with a light microscope (BZ9000, Keyence, Osaka, Japan).

## Statistical analysis

Data in Figs 1, 2, and 4 are presented as the mean ± SEM, and were analyzed by unpaired Student's $t$-tests with Welch's correction. Data in Figs 5 and 6 are presented as the mean ± SEM, and were analyzed by unpaired Student's $t$-tests. Data in Fig 3 are presented as the mean ± SEM, and were analyzed by two-way ANOVA and Bonferroni's multiple-comparisons test. All analyses were carried out with Prism 8 software (GraphPad Software, San Diego, CA, USA).

## Study approval

The patients were pregnant women who delivered at Juntendo University between March and May 2015. The total number of the patients with singleton pregnancy and full-term delivery

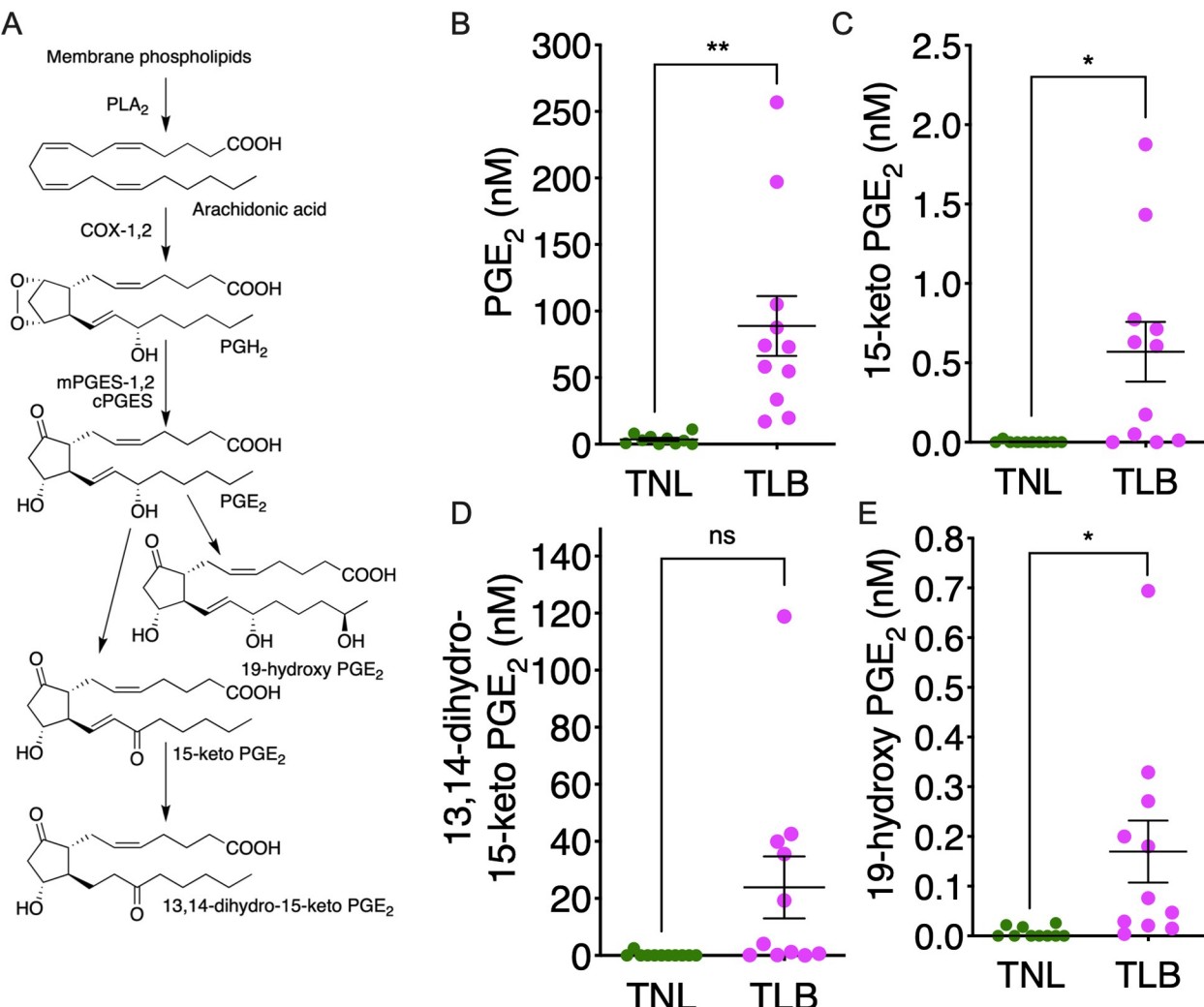

**Fig 1. Levels of prostaglandin E$_2$ (PGE$_2$) and its metabolites in amniotic fluid.** (A) PGE$_2$ biosynthesis and metabolism. Arachidonic acid released from membrane phospholipids by phospholipase A2 (PLA2) enzymes is acted on by either of the two cyclooxygenase (COX) isozymes, COX-1 or COX-2. The COX metabolite PGH$_2$ is then converted to PGE$_2$ by microsomal PGE synthase (mPGES-1), mPGES-2, or cytosolic PGE synthase (cPGES). PGE$_2$ is then oxidized by 15-hydroxyprostaglandin dehydrogenase (15-PGDH) to form 15-keto PGE$_2$, or by cytochrome P450 4F8 (CYP4F8) to form 19-hydroxy PGE$_2$. (B–E) Levels of PGE$_2$ and its metabolites in amniotic fluid from patients with term not in labor (TNL) and term in labor (TLB) deliveries were quantified by LC-MS/MS. TNL, $n = 10$; TLB, $n = 11$. Horizontal lines and error bars indicate the mean ± SEM. Data were analyzed by unpaired Student's $t$-tests with Welch's correction. $^*p < 0.05$, $^{**}p < 0.01$; 'ns', not significant.

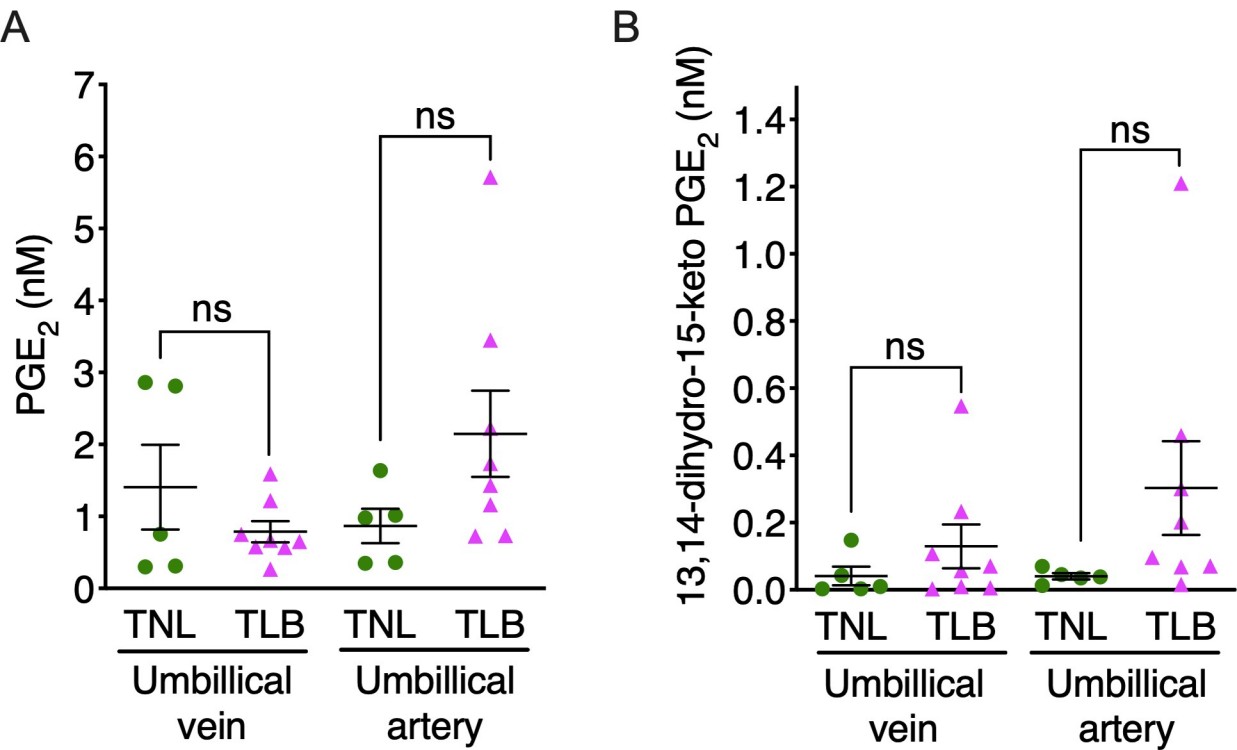

**Fig 2. Levels of prostaglandin E₂ (PGE₂) and its metabolites in umbilical cord blood.** Levels of PGE₂ and its metabolites in umbilical cord blood from patients with term not in labor (TNL) and term in labor (TLB) deliveries were quantified by LC-MS/MS. (A) PGE₂ and (B) 13,14-dihydro-15-keto-PGE₂ in blood from umbilical cord veins and arteries of patients with term not in labor (TNL) and term in labor (TLB) deliveries were quantified by LC-MS/MS. TNL, *n* = 6; TLB, *n* = 8. Horizontal lines and error bars indicate the mean ± SEM. Data were analyzed by unpaired Student's *t* tests with Welch's correction. 'ns', not significant.

was 218, and the patients with planned cesarean section was 25. Ten patients with breech presentation were recruited to the TNL group and agreed to participate in our study. The total number of the patients with spontaneous delivery was 112, and the 28 patients were excluded due to underlying disease, medication, preterm rupture of membrane, or bacterial vaginosis during pregnancy. Eleven patients were recruited the TLB group and agreed to participate in our study. The patients were all provided written informed consent prior to sample collection under a protocol approved by the Juntendo University Research Ethics Committee (13–131). All methods were performed in accordance with the Declaration of Helsinki.

## Results

### Study participants

The study included 21 pregnant women at term, 10 with TLB deliveries and 11 with TNL deliveries. Maternal characteristics are summarized in Table 1. There are no significant differences in maternal age, gestational age, or gravidity between the women in the TNL and TLB groups.

### PGE₂ and its metabolites in amniotic fluid

The biosynthetic and metabolic pathways of PGE₂ are shown in Fig 1A. Levels of PGE₂ and its metabolites in amniotic fluid from women delivering at TNL or TLB were measured by LC-MS/MS. The mean PGE₂ concentration in amniotic fluid was significantly higher in TLB

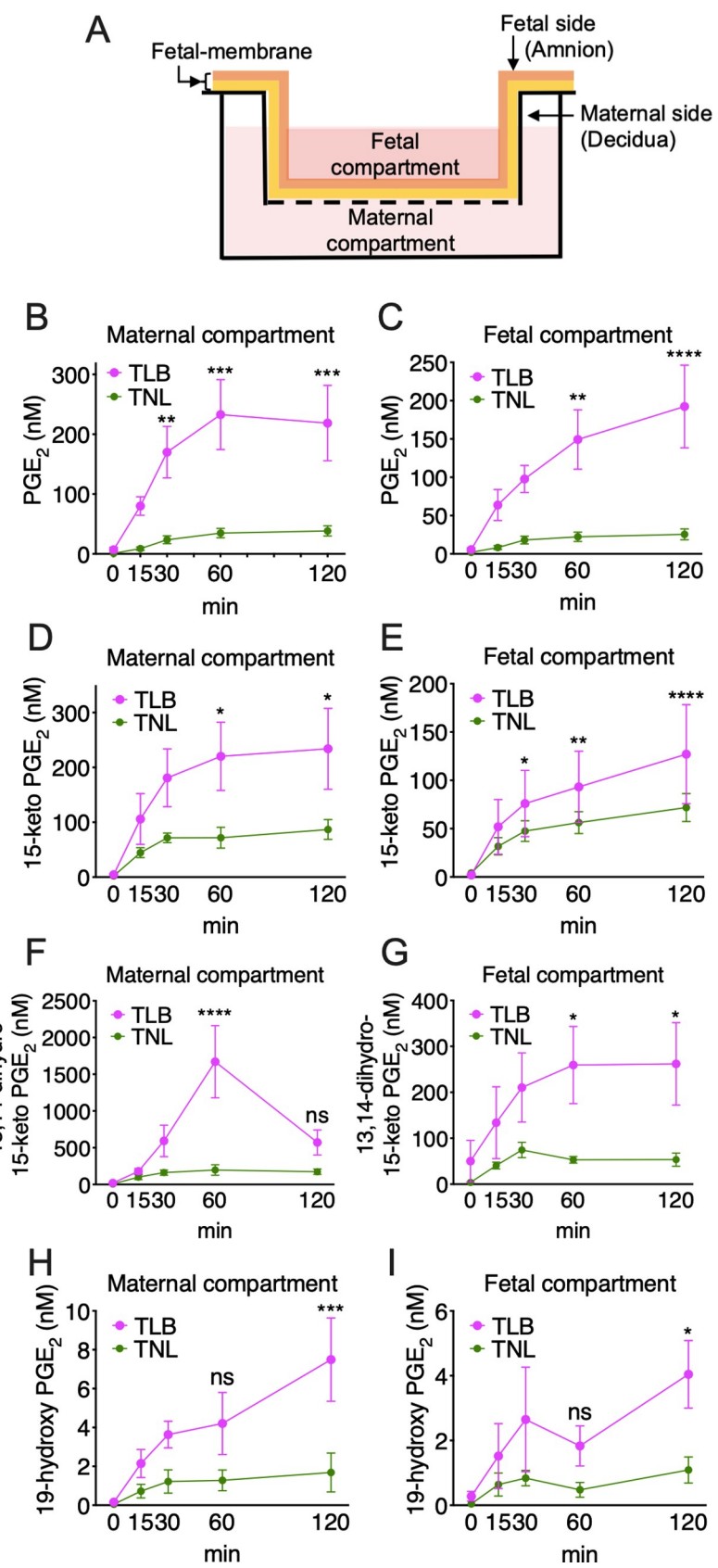

**Fig 3. Production of prostaglandin E$_2$ (PGE$_2$) and its metabolites by fetal-membrane *in vitro*.** (A) Schematic presentation of fetal-membrane Transwell assay. (B-I) Accumulation of PGE$_2$ (B and C),15-keto-PGE$_2$ (D and E), 13,14-dihydro-15-keto-PGE$_2$ (F and G) and 19-hydroxy-PGE$_2$ (H and I) in maternal and fetal compartments was quantified by LC-MS/MS. Circle markers and error bars indicate the mean ± SEM. Data were analyzed by two-way ANOVA. $^*p <$0.05, $^{**}p <$0.01, $^{***}p <$0.005, $^{****}p <$0.0001. Term not in labor (TNL) samples, $n = 6$; term in labor (TLB) samples, $n = 6$.

(89.0 nM) than in TNL (3.6 nM, $p <$0.01) (Fig 1B). Concentrations of the following PGE$_2$ metabolites in amniotic fluid were also significantly higher in TLB than in TNL: 15-keto-PGE$_2$ (0.6 nM versus 0.0 nM, $p <$0.05) (Fig 1C), 13,14-dihydro-15-keto-PGE$_2$ (23.9 nM versus 0.2 nM, $p = $0.05) (Fig 1D), and 19-hydroxy-PGE$_2$ (0.2 nM versus 0.0 nM, $p <$0.05) (Fig 1E).

## PGE$_2$ and its metabolites in umbilical-cord blood

To examine whether PGE$_2$ and its metabolites were produced by placental villous tissue, their concentrations were measured in blood plasma of the umbilical artery (UmA) and vein (UmV) by LC-MS/MS (Fig 2). In UmV plasma, the mean concentrations of PGE$_2$ (1.4 nM versus 0.8 nM in the TNL and TLB groups, respectively, $p = $0.23) (Fig 2A) and 13,14-dihydro-15-keto-PGE$_2$ (0.0 nM versus 0.1 nM, in the TNL and TLB groups, respectively, $p = $0.33) (Fig 2B) were low when compared with amniotic fluid, and did not differ significantly. Similarly, the median concentrations in UmA plasma of PGE$_2$ (0.9 nM versus 2.1 nM, $p = $0.13) (Fig 2A) and 13,14-dihydro-15-keto-PGE$_2$ (Fig 2B) (0.0 nM versus 0.3 nM, $p = $0.17) were comparable between TNL and TLB groups. As umbilical-cord blood consists of the blood flow from the placenta, these data suggest that placental villous tissue does not produce large amounts of PGE$_2$ or its metabolites.

## PGE$_2$ and its metabolites are produced by fetal-membrane

We next investigated PG production by fetal-membrane, which cover a large surface area of the uterus and placenta, with one side (the amniotic epithelium) in direct contact with the amniotic fluid. Release of PGE$_2$ from the fetal-membrane to the maternal and fetal compartments was measured in Transwell assays (Fig 3A). PGE$_2$ concentrations in both maternal (decidual) and fetal (amniotic) compartments in the TLB group were rapidly increased compared to the TNL group (Fig 3B and 3C). The concentrations of PGE$_2$ metabolites 15-keto-PGE$_2$ (maternal compartment, $p <$0.05; fetal compartment, $p = $0.65) (Fig 3D and 3E), 13,14-dihydro-15-keto-PGE$_2$ (maternal compartment, $p = $0.48; fetal compartment, $p <$0.05) (Fig 3F and 3G), and 19-hydroxy-PGE$_2$ (maternal compartment, $p <$0.001; fetal compartment, $p <$0.05) at 120 min (Fig 3H and 3I) were also higher in TLB than in TNL. These findings suggested that PGE$_2$ and its metabolites are biosynthesized in fetal-membranes and released rapidly into the amniotic fluid.

## Expression of PGE$_2$-related genes and enzymes in fetal-membrane

To clarify the enzymes related to PGE$_2$ production and its release into amniotic fluid, we performed quantitative reverse-transcription PCR (RT-qPCR) analysis of the fetal-membrane for expression of genes encoding various PGE$_2$ biosynthetic enzymes. The expression levels of *PLA2G4A* mRNA, which encodes cytosolic phospholipase A2α (cPLA2α), were significantly higher in TLB than in TNL ($p <$0.05) (Fig 4A). No significant differences were observed in the expression of *PTGS1* mRNA, which encodes COX-1 ($p = $0.13), or of *PTGS2* mRNA, which encodes COX-2 ($p = $0.07), between the TNL and TLB samples (Fig 4B and 4C). Notably, among the genes encoding the three PGES isozymes, only *PTGES3*, which encodes cPGES,

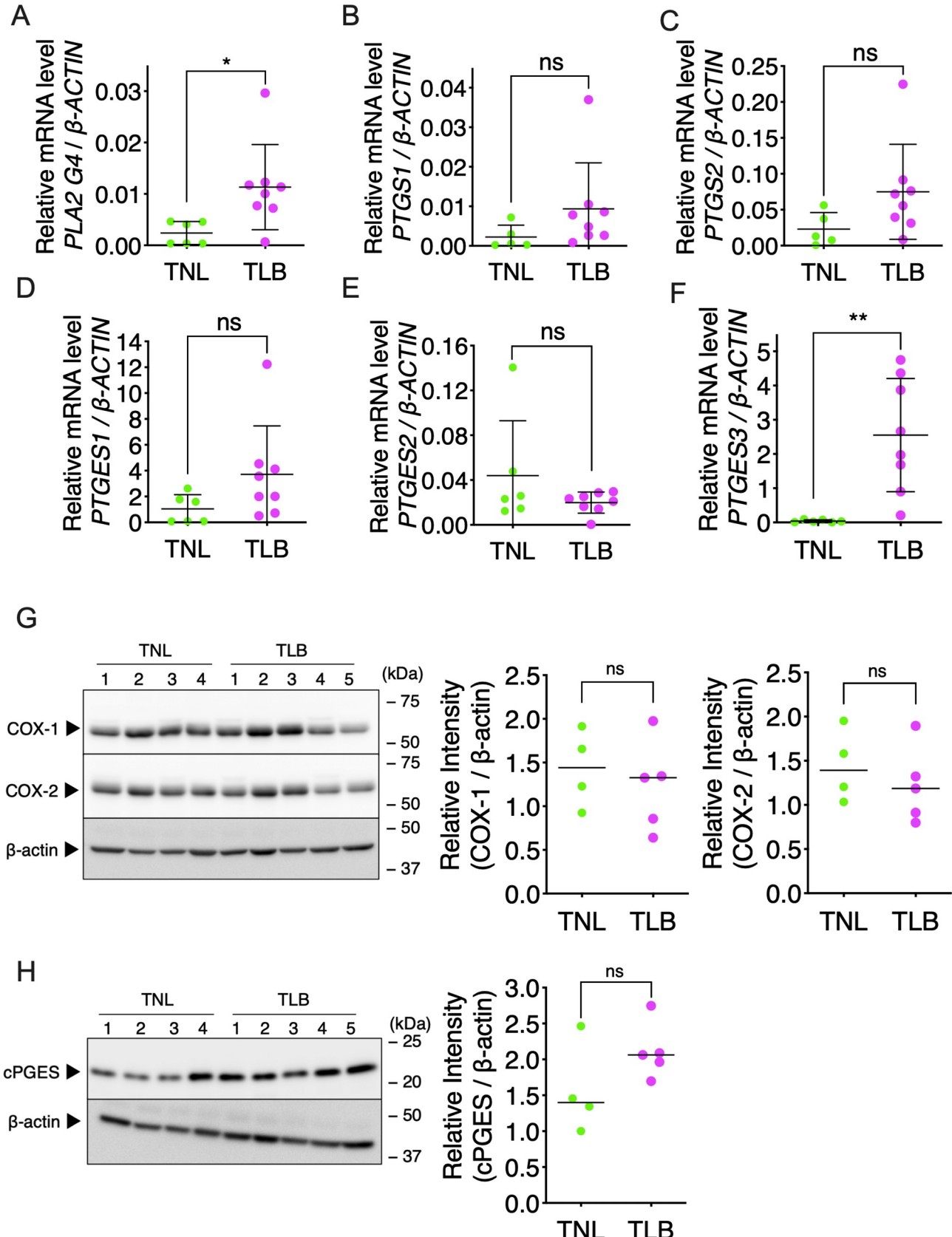

**Fig 4. Expression of genes of the prostaglandin E$_2$ (PGE$_2$) biosynthetic enzymes in fetal-membrane.** Relative levels of mRNAs encoding genes involved in PGE$_2$ biosynthesis in fetal-membrane of patients with term not in labor (TNL) or term in labor (TLB) deliveries were measured by RT-qPCR. (A) *PLA2G4A*, (B) *PTGS1*, (C) *PTGS2*, (D) *PTGES1*, (E) *PTGES2*, (F) *PTGES3*. Horizontal lines and error bars indicate the mean ± SEM. TNL, $n = 6$; TLB, $n = 8$. Western blot analysis of COX-1 and COX-2 (G), and cPGES (H) in fetal-membrane. TNL, $n = 4$; TLB, $n = 5$. Data were analyzed by unpaired Student's $t$-tests with Welch's correction. $^{*}p < 0.05$, $^{**}p < 0.01$; 'ns', not significant.

showed a significant difference, with higher expression in TLB than in TNL fetal-membrane ($p < 0.005$) (Fig 4F); by contrast, the levels of *PTGES1* ($p = 0.09$) and *PTGES2* ($p = 0.28$) mRNAs were similar in the two groups (Fig 4D and 4E). Western blotting of fetal-membrane extracts showed that COX-1 and COX-2 protein levels were similar between TNL and TLB (Fig 4G), however, cPGES protein levels were relatively higher in TLB than in TNL (Fig 4H). Although the patient of TNL1, 2, and 3 had no operation before pregnancy, the patient of TNL4 had the uterine operation before pregnancy. The higher expression of cPGES in TNL4 might be due to the uterine operation.

To precisely determine the site of PGE$_2$ production, the fetal-membranes of 8 patients in TNL and 7 patients in TLB were separated into the amnion and the chorion, and the expression of PGE$_2$-related genes was compared between TNL and TLB. There were no significant differences between the women in the TNL and TLB groups with respect to the following variables: maternal age (34.0 years versus 34.5 years, respectively, $p = 0.74$), gestational age ($38^{6/7}$ weeks versus $39^{1/7}$ weeks, $p = 0.37$), gravidity (0.0 versus 0.0, $p > 0.99$), or parity (0.0 versus 0.0, $p = 0.44$). ($p$-values of maternal age and gestational age were determined by unpaired Student's $t$-tests, and $p$-values of gravidity and parity were determined by Mann-Whitney $U$ test.) The mRNA expression of chorionic *PLA2G4A*, amniotic *PTGS2*, and amniotic *PTGES3* was significantly higher in the TLB than in the TNL ($p < 0.01$, $p < 0.005$, and $p < 0.05$, respectively) (Fig 5A, 5C, and 5F) The levels of *PTGS1*, *PTGES1*, and *PTGES2* mRNAs were similar on both the chorion and amnion of the TNL and TLB samples (Fig 5B, 5D, and 5E). These data suggest that PGE$_2$ is preferably produced from the amnion of fetal-membrane.

## Localization of COX-1, COX-2, and cPGES in fetal-membrane

To identify PGE$_2$-producing cells, fetal-membrane were immunohistochemically stained with antibodies to COX-1, COX-2, and cPGES. Anti-cPGES antibody was also used for western blotting (Fig 4G). These enzymes were mainly expressed in chorionic trophoblasts, mesenchymal cells, and amniotic epithelium (Fig 6). These results suggest that COX-1 and/or COX-2, and cPGES in these cells are involved in the production of PGE$_2$.

## Expression and localization of PG transporter in fetal-membrane

PGE$_2$ is transported to the extracellular space and exerts its effects by binding to target receptors [15]. The PG transporter SLCO2A1 transports several substrates including PGE$_2$ [16]. To investigate the expression of SLCO2A1 in chorionic trophoblasts, mesenchymal cells, and amniotic epithelium, we performed RT-qPCR analysis and immunohistochemistry (Fig 7). *SLCO2A1* mRNA was detected in fetal-membrane, but its levels of expression did not differ between TNL and TLB (Fig 7A). Similarly, immunohistochemical analysis of SLCO2A1 in fetal-membrane revealed similar staining in chorionic trophoblasts, mesenchymal cells, and amniotic epithelium in TNL and TLB samples (Fig 7B and 7C). These results suggest that cPGES, and SLCO2A1 in chorionic trophoblasts, mesenchymal cells, and amniotic epithelium are involved in the production and release of PGE$_2$.

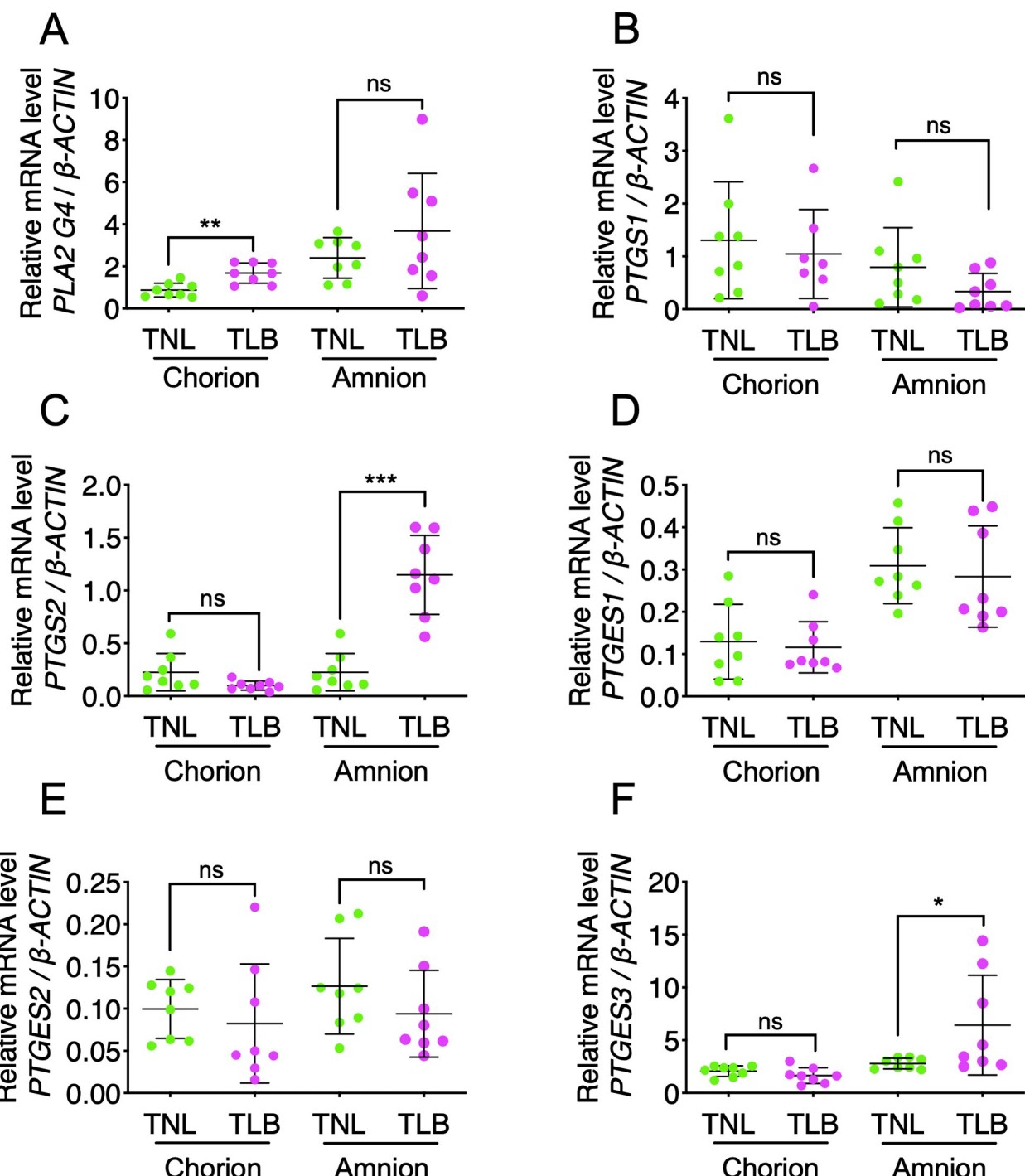

**Fig 5. Expression of genes of the prostaglandin E$_2$ (PGE$_2$) biosynthetic enzymes in the amnion and the chorion.** Relative levels of mRNAs encoding genes involved in PGE$_2$ biosynthesis in the amnion and the chorion of patients with term not in labor (TNL) or term in labor (TLB) deliveries were measured by RT-qPCR. (A) *PLA2G4A*, (B) *PTGS1*, (C) *PTGS2*, (D) *PTGES1*, (E) *PTGES2*, (F) *PTGES3*. Horizontal lines and error bars indicate the mean ± SEM. TNL, *n* = 8; TLB, *n* = 8. Data were analyzed by unpaired Student's *t*-tests. *$p < 0.05$, **$p < 0.01$, ***$p < 0.005$; 'ns', not significant.

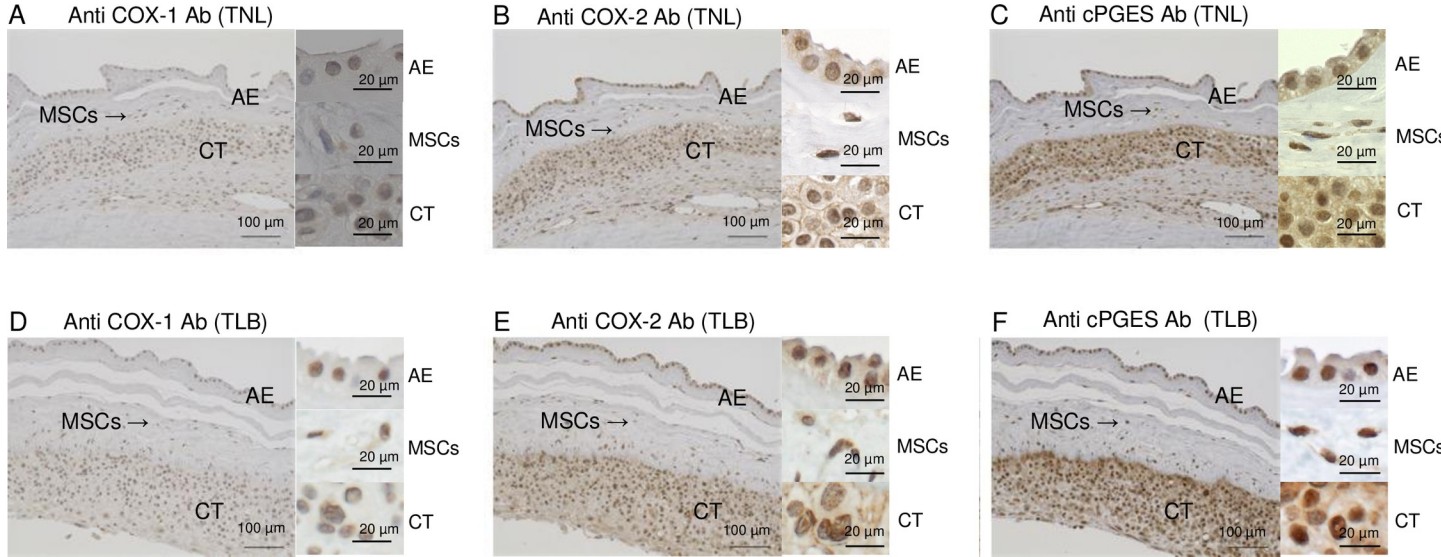

**Fig 6. Immunohistochemical analyses of prostaglandin E$_2$ (PGE$_2$) biosynthetic enzymes proteins in fetal-membrane.** Immunohistochemical staining of fetal-membrane with antibodies (Ab) specific for cyclooxygenase-1 (COX-1) (A, D), cyclooxygenase-2 (COX-2) (B, E), and cytosolic PGE synthase (cPGES) (C, F). Hematoxylin was used for nuclear counterstaining. Membrane sections were from patients with term not in labor (TNL; A–C) and term in labor (TLB; D–F) deliveries. Scale bars = 100 μm (Left; low-power field), Scale bars = 20 μm (Right: high-power field). AE, amniotic epithelium; MSCs, mesenchymal cells; CT, chorionic trophoblast.

## Discussion

PGE$_2$ is known to play an important role in human labor. Preventing the PGE$_2$ biosynthesis by administration of a nonselective cyclooxygenase (COX) inhibitor effectively delays parturition in mice [17]. Celecoxib, a selective inhibitor of COX-2, can also delay human preterm labor [18]. Vaginal administration of PGE$_2$ tablets at term increases the likelihood of a transvaginal delivery within 24 h [19], and vaginal and cervical application of PGE$_2$ is more effective than intravenous injection of oxytocin for induction of vaginal delivery within 24 h [20]. The administration of PGE$_2$ for labor induction is permitted by the Japan Society of Obstetrics and Gynecology guidelines and is commonly used in the clinical situation in Japan [21].

Recently, PGs and their metabolites were shown to accumulate in amniotic fluid in TLB deliveries [3], but the detailed molecular mechanism underlying PGs production during human spontaneous labor remains unclear. To determine the enzymes and transporter(s) responsible for the production and accumulation of PGE$_2$ in the amniotic fluid, we conducted the current study. Firstly, we confirmed that high levels of PGE$_2$ and its metabolites accumulate in the amniotic fluid of TLB, but not in that of TNL. The PGE$_2$ concentration was much

**Table 1. Maternal characteristics of the study participants.**

|  | TNL (*n* = 10) | TLB (*n* = 11) | *p*-value |
|---|---|---|---|
| Maternal age (years) | 36.3 (27–42) | 34.9 (30–40) | 0.39 |
| Gestational age at delivery (weeks) | 38$^{3/7}$ (38–39) | 38$^{5/7}$ (38–39) | 0.13 |
| Gravidity | 1.0 (0–1) | 0.5 (0–2) | 0.85 |
| Parity | 0.0 (0–1) | 0.0 (0–1) | 0.64 |

Values are expressed as the median (full range). *p*-values of maternal age and gestational age were determined by unpaired Student's *t*-tests, and *p*-values of gravidity and parity were determined by Mann-Whitney *U* test. TNL, term not in labor; TLB, term in labor.

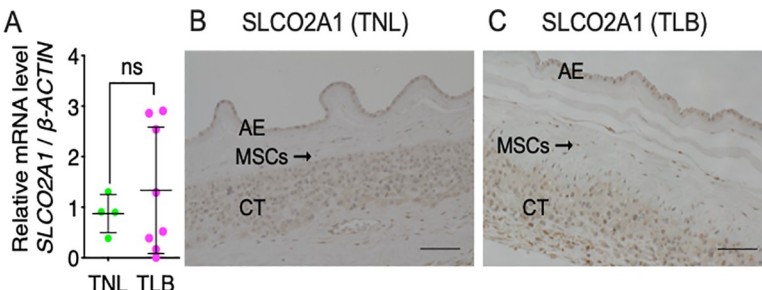

**Fig 7. Expression of the prostaglandin transporter in fetal-membrane.** (A) Relative levels of mRNA encoding *SLCO2A1* in fetal-membrane of patients with term not in labor (TNL) or term in labor (TLB) deliveries were measured by RT-qPCR. Horizontal lines and error bars indicate the mean ± SEM. Data were analyzed by unpaired Student's *t*-tests. 'ns', not significant. TNL, $n = 4$; TLB, $n = 8$. (B, C) Immunohistochemical staining of fetal-membrane from patients with (B) TNL and (C) TLB deliveries with an anti-SLCO2A1 antibody. Hematoxylin was used for nuclear counterstaining. Scale bars = 100 μm. AE, amnion epithelium; MSCs, mesenchymal cells; CT, chorion trophoblast.

higher in amniotic fluid than in umbilical cord blood, suggesting the importance of amniotic $PGE_2$ in inducing labor. It was previously reported that the major site of PG synthesis in human pregnancy is fetal-membrane [2]. The fetal-membrane, which consists of the chorion and amnion, is a major source of amniotic fluid [22]. The amnion consists of a single layer of epithelial cells and subepithelial mesenchymal layer, and directly contacts with amniotic fluid. The chorion consists of trophoblast cells and decidua, and directly contacts with the maternal myometrium. In agreement with previous reports [23,24], we confirmed that the level of $PGE_2$ produced from fetal-membrane was significantly higher in TLB than TNL. RT-PCR analysis (Fig 5) and immunohistochemical staining (Figs 6 and 7) suggest that both amnion and the chorion are able to produce and release $PGE_2$. Because the amnion directly contacts amniotic fluid, it may contribute to $PGE_2$ accumulation amniotic fluid during labor. There was no correlation between the amniotic $PGE_2$ concentration and the length of labor or the number of examinations. We also showed that SLCO2A1 is expressed in term fetal-membrane, which is consistent with the previous report [15]. 15-hydroxyprostaglandin dehydrogenase (PGDH) is an important enzyme in PG inactivation, and it converts $PGE_2$ into its inactive form, 15-keto $PGE_2$ [25]. Because PGDH is not expressed in the amniotic epithelium at term [26], intact $PGE_2$ escapes from inactivation and accumulates in amniotic fluid.

$PGE_2$ induces uterine contractions by binding to G-protein-coupled $PGE_2$ receptors (EPs) expressed in the uterine myometrium [27]. Four EPs bind $PGE_2$, with $K_d$ values ranging from 1 nM to 40 nM [28]; EP1 and EP3 are highly expressed in human myometrium at term [29]. We identified a median concentration of $PGE_2$ in amniotic fluid in the TLB group of ~70 nM, which is sufficient to activate these EP receptors. The fetal-membrane Transwell assay showed both fetal and maternal side of the TLB fetal-membrane produced a high concentration of $PGE_2$. As the chorion exposes to the uterine myometrium, EPs in uterine myometrium may be activated by the accumulated $PGE_2$ in amniotic fluid.

Multiple enzymic reactions are required for $PGE_2$ biosynthesis from AA. $cPLA_2$s are a group of enzymes that hydrolyze the phospholipids to release fatty acids including AA [30]. Cyclooxygenases are responsible for the generation of the PG precursor $PGH_2$ and indispensable for $PGE_2$ production [31]. In agreement with the previous report [32], immunohistochemistry showed the expression of COX-1, COX-2, and cPGES in the amniotic epithelium, mesenchymal cells in the amnion, and chorionic trophoblasts in the chorion. Some studies suggested that the $PGE_2$ biosynthesis in the fetal-membrane is mediated by an increase of *PTGS2* mRNA expression in amnion cells [33]. Among the three $PGE_2$ synthases, cPGES is

less well understood. cPGES was originally purified from the cytosolic fraction of lipopolysaccharide-treated rat brain and has been shown to be ubiquitously expressed [34]. Although its roles *in vivo* remain unclear, cPGES-deficient mice die before birth (possibly as a result of lung aplasia), with abnormal skin and lung morphology with reduced $PGE_2$ content in the lungs compared with wild-type mice [35]. By contrast, mPGES-1 deficient mice [36] and mPGES-2 deficient mice [37] develop and deliver normally. Taken together, these results suggest that cPGES contributes to $PGE_2$ production in fetal-membrane in human labor.

In clinical practice, uterine contractions in patients with weak labor are frequently augmented by artificial rupture of the amniotic membrane (ROM) [38]. Previous studies have reported the relationship between the release of PGs at amniotomy and the subsequent onset of labor [39,40]. The relationship between ROM/amniotomy and augmentation of labor could be explained by the high level of $PGE_2$ in amniotic fluid. In addition, extra-amniotic $PGE_2$ administration can also achieve a 25% reduction in the length of induced labor [41]. This extra-amniotic $PGE_2$ might also enhance myometrial contractions following SLCO2A1-mediated transport across the membranes of myometrium.

The major limitation in our study is that the fetal-membrane was taken after labor, and we could not exclude the possibility that $PGE_2$ was produced after labor. Another limitation is that the amniocentesis was performed transvaginally from the TLB group, but that was performed transabdominally from TNL. Because the transabdominal amniocentesis may induce serious complications, we could not collect the amniotic fluid during pregnancy. Thus, we could not exclude the effects of cervicovaginal secretions and bacteria in TLB subjects. To minimize these effects, we washed the vagina with the saline and wiped it with sterilized gauze. In addition, we excluded the patients who were positive for the cultured examination of the vaginal swab. In summary, our results suggest that cPGES, and SLCO2A1 contribute to $PGE_2$ production from fetal-membrane in labor.

## Supporting information

**S1 Fig. The raw images of Fig 4G.**
(TIF)

## Author Contributions

**Conceptualization:** Nanase Takahashi, Toshiaki Okuno, Shintaro Makino, Takehiko Yokomizo.

**Funding acquisition:** Kazuko Saeki, Takehiko Yokomizo.

**Investigation:** Nanase Takahashi, Hiroki Fujii, Masaya Takahashi, Mai Ohba.

**Methodology:** Mai Ohba.

**Project administration:** Toshiaki Okuno.

**Resources:** Shintaro Makino, Masaya Takahashi, Kazuko Saeki.

**Supervision:** Atsuo Itakura, Satoru Takeda, Takehiko Yokomizo.

**Validation:** Shintaro Makino.

**Writing – original draft:** Nanase Takahashi, Toshiaki Okuno.

**Writing – review & editing:** Nanase Takahashi, Toshiaki Okuno, Takehiko Yokomizo.

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
