## [Decision Letter · Decision Letter 0]

10 Nov 2020

PONE-D-20-30000

Up-regulation of cytosolic prostaglandin E synthase in fetal-membrane and amniotic prostaglandin E2 accumulation in labor

PLOS ONE

Dear Dr. Okuno,

Thank you for submitting your manuscript to PLOS ONE. After careful consideration, we feel that it does not fully meet PLOS ONE’s publication criteria as it currently stands. Therefore, we invite you to submit a revised version of the manuscript that addresses the points raised during the review process. The reviewers have raised major concerns on the quality of the data and the reliability of your conclusion.

We look forward to receiving your revised manuscript.

Kind regards,

Hai-Yan Lin

Academic Editor

PLOS ONE

Journal Requirements:

2. In the Methods section, please provide the sequences of the specific primers used in the RT-qPCR assays conducted in your study.

3. In your Methods section, please provide additional information about the participant recruitment method and the demographic details of your participants. Please ensure you have provided sufficient details to replicate the analyses such as: a) the recruitment date range (month and year), b) a description of how participants were recruited.

Reviewers' comments:

Reviewer's Responses to Questions

**Comments to the Author**

1. Is the manuscript technically sound, and do the data support the conclusions?

Reviewer #1: Partly

Reviewer #2: No

2. Has the statistical analysis been performed appropriately and rigorously? 

Reviewer #1: Yes

Reviewer #2: No

3. Have the authors made all data underlying the findings in their manuscript fully available?

Reviewer #1: Yes

Reviewer #2: Yes

4. Is the manuscript presented in an intelligible fashion and written in standard English?

Reviewer #1: Yes

Reviewer #2: Yes

5. Review Comments to the Author

Reviewer #1: The authors describe the upregulation of cPGES upon labor. While the work presented by the authors represents an interesting perspective, some major concerns need to be addressed:

1) Although the authors observed no significant differences of mRNAs encoding COX-1 or COX-2, that does not necessarily mean the protein level would be the same pattern. Since they did western blot to check the expression of cPGES protein in fetal-membranes, did they also check COX-1 or COX-2 protein as well?

2) Simply by the weak staining of COX-1 in fetal-membranes are not sufficient to eliminate its role in PGE2 production. The weak staining could either be due to weak expression or, most likely, the antibody issue. Even different antibodies targeting the same antigen could result in highly variable staining intensity, so it’s not possible to directly compare the staining results from different antibodies targeting different antigens. The authors need to rephrase some statements they made correspondingly.

3) The authors mentioned that an inhibitor of COX-2 could delay human preterm labor. While they observed the mRNA encoding COX-2 was higher in TLB in the amnion, and cPGES protein expression was higher in TLB in the fetal membrane. The authors need to discuss a bit more about the role of these different proteins in different places.

Minors:

Figure 6: the current magnification makes it difficult to tell if the staining of cPGES was more intense in a particular group. The authors could include a higher magnification from where the positive signals are for better displaying the results.

Reviewer #2: This work studied prostaglandin (PG) biosynthetic enzymes and transporter in the accumulation of PGE2 in amniotic fluid during human labor. Consistent with prior literature, PGE2 and its metabolites were increased in amniotic fluid from women in labor. Using fetal membrane transwell assays, levels of PGE2 in both maternal and fetal compartments were increased in the labor relative to not in labor. Cytosolic PGE synthase 3 and COX-2 mRNA and protein were increased significantly in labor relative to before labor with increases in the transported protein SLCO2A1 expressed in chorionic trophoblast and amniotic epithelium.

Overall, this study and its conclusions are scientifically flawed. First, samples were collected after labor – therefore it cannot be concluded that amniotic fluid PGE2 initiates labor because it may occur as a result of labor. Please change all wording regarding this conclusion throughout including the abstract. Second, it should be emphasized that fluid was collected from the “not in labor” group by amniocentesis and the time of cesarean section and is thereby free of cervical, vaginal, and myometrial-derived PGE2. In contrast, amniotic fluid from women in labor was collected at the time of vaginal delivery with ruptured membranes and is thereby contaminated with cervico-vaginal secretions and bacteria. There is a wide rage of PGE2 levels in fluid from women in labor. Hence, there is unequal variance and cannot be analyzed by Student’s t test. Please correlate the amount of PGE2 in fluid with length of labor, number of exams, and meconium or not. One patient, in particular, is outside the normal range suggesting other mechanisms in play including subclinical infection or other contaminants.

3, The transwell assay was conducted with a large section of fetal membrane (5x5 cm). Is this an error? The contents of fetal compartment vs maternal compartment is not explained.

4. Gravidity and parity cannot be analyzed by student’s t test.

5. Immunoblot in figure 5G clearly indicates that the normalizer protein (beta actin) is not consistently expressed with at least 2 samples in the TNL group increased relative to the lower levels of cPGES. This generates and low levels of cPGES relative to beta actin in 2 of 4 samples. The comparison is thereby not valid and I do not believe the conclusion that cPGES is increased in labor.

Minor:

1. Abstract: encodes cytosolic prostaglandin E synthase 3

6. PLOS authors have the option to publish the peer review history of their article (what does this mean?). If published, this will include your full peer review and any attached files.

Reviewer #1: No

Reviewer #2: No

---

## [Author Response · Author response to Decision Letter 0]

30 Mar 2021

March 30, 2021

Dr. Hai-Yan Lin

Academic Editor

PLOS One

Dear Editor, 

Please find enclosed our revised manuscript entitled “Up-regulation of cytosolic prostaglandin E synthase in fetal-membrane and amniotic prostaglandin E2 accumulation in labor.” (PONE-D-20-30000).

We thank you and the two reviewers for valuable comments and useful suggestions. We have addressed the comments raised by the reviewers, and the amendments are highlighted in red in the revised manuscript. The blot/gel image data are added in Supporting Information. In addition, we made an inquiry to Thermo Fisher Scientific about the sequences of the primers for quantitative PCR, however, they mentioned that the sequences are undisclosed and the assay IDs should be shown in the manuscript. We also added a new author, Hiroki Fujii, because he performed additional western blot analyses.

We hope that these revisions meet the reviewers’ requests. Point-by-point responses to the reviewers’ comments are listed below this letter.

We look forward to hearing from you at your earliest convenience.

Sincerely,

Toshiaki Okuno Ph.D.

Juntendo University School of Medicine

Department of Biochemistry

Hongo 2-1-1, Bunkyo-ku, Tokyo 113-8421, Japan

Tel: 81-3-5802-1031

Fax: 81-3-5802-5889

Email Address: tokuno@juntendo.ac.jp

Point-by-point Responses

Reviewer #1: The authors describe the upregulation of cPGES upon labor. While the work presented by the authors represents an interesting perspective, some major concerns need to be addressed: 

Response: We thank the reviewer for his/her comments that helped to improve the manuscript.

1)Although the authors observed no significant differences of mRNAs encoding COX-1 or COX-2, that does not necessarily mean the protein level would be the same pattern. Since they did western blot to check the expression of cPGES protein in fetal-membrane, did they also check COX-1 or COX-2 protein as well? 

Response: Thank you for your critical comments. We agree with you and performed western blot analysis of COX-1 and COX-2. The representative results are shown in New Figure 4G, and we have revised the manuscript as follows.

P2L27: Western blot analyses revealed that the levels of COX-1 and COX-2 were comparable between the two groups, however, the level of cPGES was relatively higher in TLB than in TNL.

P20L294: Western blotting of fetal-membrane extracts showed that COX-1 and COX-2 protein levels were similar between TNL and TLB (Fig 4G), however, cPGES protein levels were relatively higher in TLB than in TNL (Fig 4H).

2)Simply by the weak staining of COX-1 in fetal-membrane are not sufficient to eliminate its role in PGE2 production. The weak staining could either be due to weak expression or, most likely, the antibody issue. Even different antibodies targeting the same antigen could result in highly variable staining intensity, so it’s not possible to directly compare the staining results from different antibodies targeting different antigens. The authors need to rephrase some statements they made correspondingly.

Response: Thank you for your critical comment. We agree with your opinion that the weak staining is not sufficient to eliminate the roles of the protein and have revised the manuscript as follows. 

P23L337: We deleted the sentence “Although COX-1 staining was weak in these tissues in both TNL and TLB samples (Fig 6A and D), staining of COX-2 (Fig 6B and E) and cPGES (Fig 6C and F) was strong. Notably, staining of cPGES was more intense in TLB samples than in TNL (Fig 6C and F).”

P23L337: These results suggest that COX-1 and/or COX-2, and cPGES in these cells are involved in the production of PGE2.

3) The authors mentioned that an inhibitor of COX-2 could delay human preterm labor. While they observed the mRNA encoding COX-2 was higher in TLB in the amnion, and cPGES protein expression was higher in TLB in the fetal membrane. The authors need to discuss a bit more about the role of these different proteins in different places.

Response: Thank you for your useful comment. In this manuscript, we define that the fetal-membrane comprises the amnion, chorion, and decidua. Therefore, we conclude that both COX-2 and cPGES were expressed in the same cells in the amnion of the fetal-membrane, and we have revised the manuscript as follows.

P28L414: In agreement with the previous report [32], immunohistochemistry showed the expression of COX-1, COX-2, and cPGES in the amniotic epithelium, mesenchymal cells in the amnion, and chorionic trophoblasts in the chorion.

Minors:

Figure 6: the current magnification makes it difficult to tell if the staining of cPGES was more intense in a particular group. The authors could include a higher magnification from where the positive signals are for better displaying the results.

Response: Thank you very much for your useful comments. We have included higher magnification figures in New Figure 6.

We appreciate Reviewer #1 for the constructive and insightful comments, which helped us to improve our manuscript. 

Reviewer #2: This work studied prostaglandin (PG) biosynthetic enzymes and transporter in the accumulation of PGE2 in amniotic fluid during human labor. Consistent with prior literature, PGE2 and its metabolites were increased in amniotic fluid from women in labor. Using fetal membrane transwell assays, levels of PGE2 in both maternal and fetal compartments were increased in the labor relative to not in labor. Cytosolic PGE synthase 3 and COX-2 mRNA and protein were increased significantly in labor relative to before labor with increases in the transported protein SLCO2A1 expressed in chorionic trophoblast and amniotic epithelium.

Overall, this study and its conclusions are scientifically flawed. First, samples were collected after labor – therefore it cannot be concluded that amniotic fluid PGE2 initiates labor because it may occur as a result of labor. Please change all wording regarding this conclusion throughout including the abstract. Second, it should be emphasized that fluid was collected from the “not in labor” group by amniocentesis and the time of cesarean section and is thereby free of cervical, vaginal, and myometrial-derived PGE2. In contrast, amniotic fluid from women in labor was collected at the time of vaginal delivery with ruptured membranes and is thereby contaminated with cervico-vaginal secretions and bacteria.

Response: Thank you very much for your critical comments. We agree with you and have incorporated your suggestions throughout our manuscript.

We removed [onset of] (P2L18), [and induces labor in humans] (P3L32), [during and/or prior to labor] (P19L272), [These results suggest that cPGES is the major enzyme involved in PGE2 production during and/or prior to labor] (P21L299), [and induces labor in humans] (P30L443), and we hope that these revisions made the manuscript scientifically correct. 

We have incorporated your comments by changing the content on P2L20 [during human labor] to [in human labor] and have added new paragraphs to clarify the participants and limitations in our study as follows.

P13L195: The patients were pregnant women who delivered at Juntendo University between March and May 2015. The total number of the patients with singleton pregnancy and full-term delivery was 218, and the patients with planned cesarean section was 25. Ten patients with breech presentation were recruited to the TNL group and agreed to participate in our study. The total number of the patients with spontaneous delivery was 112, and the 28 patients were excluded due to underlying disease, medication, preterm rupture of membrane, or bacterial vaginosis during pregnancy. Eleven patients were recruited the TLB group and agreed to participate in our study.

P29L434: The major limitation in our study is that the fetal-membrane was taken after labor, and we could not exclude the possibility that PGE2 was produced after labor. Another limitation is that the amniocentesis was performed transvaginally from the TLB group, but that was performed transabdominally from TNL. Because the transabdominal amniocentesis may induce serious complications, we could not collect the amniotic fluid during pregnancy. Thus, we could not exclude the effects of cervicovaginal secretions and bacteria in TLB subjects. To minimize these effects, we washed the vagina with the saline and wiped it with sterilized gauze. In addition, we excluded the patients who were positive for the cultured examination of the vaginal swab. 

There is a wide rage of PGE2 levels in fluid from women in labor. Hence, there is unequal variance and cannot be analyzed by Student’s t test. 

Response: We agree with your assessment. We have reanalyzed the PGE2 levels in Fig. 1 and 2, and the mRNA levels in Fig. 4 by Welch’s t-test and have revised the manuscript as follows. 

P13L187: Data in Figs 1, 2, and 4 are presented as the mean ± SEM, and were analyzed by unpaired Student’s t-tests with Welch’s correction. 

P16L221: The mean PGE2 concentration in amniotic fluid was significantly higher in TLB (89.0 nM) than in TNL (3.6 nM, p <0.01) (Fig 1B). Concentrations of the following PGE2 metabolites in amniotic fluid were also significantly higher in TLB than in TNL: 15-keto-PGE2 (0.6 nM versus 0.0 nM, p <0.05) (Fig 1C), 13,14-dihydro-15-keto-PGE2 (23.9 nM versus 0.2 nM, p <0.05) (Fig 1D), and 19-hydroxy-PGE2 (0.2 nM versus 0.0 nM, p <0.05) (Fig 1E).

P17L238: Data were analyzed by unpaired Student’s t-tests with Welch’s correction.

P17L243: In UmV plasma, the mean concentrations of PGE2 (1.4 nM versus 0.8 nM in the TNL and TLB groups, respectively, p = 0.23) (Fig 2A) and 13,14-dihydro-15-keto-PGE2 (0.0 nM versus 0.1 nM, in the TNL and TLB groups, respectively, p = 0.33) (Fig 2B) were low when compared with amniotic fluid, and did not differ significantly. Similarly, the median concentrations in UmA plasma of PGE2 (0.9 nM versus 2.1 nM, p = 0.13) (Fig 2A) and 13,14-dihydro-15-keto-PGE2 (Fig 2B) (0.0 nM versus 0.3 nM, p = 0.17) were comparable between TNL and TLB groups.

P18L258: Data were analyzed by unpaired Student’s t tests with Welch’s correction. 

P20L288: No significant differences were observed in the expression of PTGS1 mRNA, which encodes COX-1 (p = 0.13), or of PTGS2 mRNA, which encodes COX-2 (p = 0.07), between the TNL and TLB samples (Fig 4B and C). Notably, among the genes encoding the three PGES isozymes, only PTGES3, which encodes cPGES, showed a significant difference, with higher expression in TLB than in TNL fetal-membrane (p <0.005) (Fig 4F); by contrast, the levels of PTGES1 (p = 0.09) and PTGES2 (p = 0.28) mRNAs were similar in the two groups (Fig 4D and E).

P22L321: Data were analyzed by unpaired Student’s t-tests with Welch’s correction.

Please correlate the amount of PGE2 in fluid with length of labor, number of exams, and meconium or not.

Response: Thank you for providing these insights. There was no patient with meconium in our study. There was no correlation between the amount of PGE2 in fluid and the length of labor or number of exams, and we have revised the manuscript as follows.

P5L65: there were no participants who had meconium in our study.

P27L396: There was no correlation between the amniotic PGE2 concentration and the length of labor or the number of examinations.

One patient, in particular, is outside the normal range suggesting other mechanisms in play including subclinical infection or other contaminants. 

Response: Thank you for your critical comment. We excluded the patient with bacterial vaginosis or chorioamnionitis, and we have revised the manuscript as follows.

P5L65: The participants with bacterial vaginosis were also excluded, and there were no participants who had meconium in our study.

3, The transwell assay was conducted with a large section of fetal membrane (5x5 cm). Is this an error? The contents of fetal compartment vs maternal compartment is not explained.

Response: Thank you for your critical comment. We have added a new Figure 3A which illustrates the fetal membrane Transwell assay and the maternal and fetal compartments. We have revised the manuscript as follows:

P8L122: Briefly, a 3 × 3 cm square of fetal-membrane was placed in a 24 mm Transwell clear insert (Corning, Lindfield, Australia) in 6-well culture plate containing serum-free culture medium (DMEM/Ham’s Nutrient Mixture F-12, phenol red-free, supplemented with 15 mM HEPES, pH 7.3 (Sigma-Aldrich, St Louis, MO, USA) and 0.5% fatty acid-free BSA (Sigma-Aldrich)). The maternal compartment on the decidua contained 3 ml medium and the fetal compartment on the amnion contained 2.5 ml medium.

P19L276: (A) Schematic presentation of fetal-membrane Transwell assay.

P19L265: PGE2 concentrations in both maternal (decidual) and fetal (amniotic) compartments in the TLB group were rapidly increased compared to the TNL group (Fig 3B and C). The concentrations of PGE2 metabolites 15-keto-PGE2 (maternal compartment, p <0.05; fetal compartment, p = 0.65) (Fig 3D and E), 13,14-dihydro-15-keto-PGE2 (maternal compartment, p = 0.48; fetal compartment, p <0.05) (Fig 3F and G), and 19-hydroxy-PGE2 (maternal compartment, p <0.001; fetal compartment, p <0.05) at 120 min (Fig 3H and I) were also higher in TLB than in TNL.

4. Gravidity and parity cannot be analyzed by student’s t test.

Response: Thank you for your critical comment. We have reanalyzed the gravidity and parity by Mann-Whiney U test (New Table 1). 

P15L215: and p-values of gravidity and parity were determined by Mann-Whitney U test.

P21L305: gravidity (0.0 versus 0.0, p > 0.99), or parity (0.0 versus 0.0, p = 0.44). (p-values of maternal age and gestational age were determined by unpaired Student’s t-tests, and p-values of gravidity and parity were determined by Mann-Whitney U test.)

5. Immunoblot in figure 5G clearly indicates that the normalizer protein (beta actin) is not consistently expressed with at least 2 samples in the TNL group increased relative to the lower levels of cPGES. This generates and low levels of cPGES relative to beta actin in 2 of 4 samples. The comparison is thereby not valid and I do not believe the conclusion that cPGES is increased in labor.

Response: Thank you for your critical comments. We have repeated western blot analysis of cPGES at least three times following protein quantification. While the intensities of beta actin were similar among all samples, the intensities of cPGES were relatively higher in TLB than in TNL (New Figure 4H). We have revised the manuscript as follows:

P2L27: Western blot analyses revealed that the levels of COX-1 and COX-2 were comparable between the two groups, however, the level of cPGES was relatively higher in TLB than in TNL.

P20L294: Western blotting of fetal-membrane extracts showed that COX-1 and COX-2 protein levels were similar between TNL and TLB (Fig 4G), however, cPGES protein levels were relatively higher in TLB than in TNL (Fig 4H). Although the patient of TNL1, 2, and 3 had no operation before pregnancy, the patient of TNL4 had the uterine operation before pregnancy. The higher expression of cPGES in TNL4 might be due to the uterine operation.

Minor:

1. Abstract: encodes cytosolic prostaglandin E synthase 3

Response: Thank you for your useful comment. There are three prostaglandin E synthase isozymes, microsomal prostaglandin E synthase (mPGES)-1, mPGES-2, and cytosolic prostaglandin E synthase (cPGES) in human. We believe that PTGES3 encodes cytosolic prostaglandin E synthase (cPGES). 

We would like to thank Reviewer #2 for taking the time and effort necessary to review the manuscript. We sincerely appreciate all your valuable comments and suggestions, which helped us to improve the quality of the manuscript. We hope that our revised manuscript will fully meet your requests.

---

## [Decision Letter · Decision Letter 1]

12 Apr 2021

Up-regulation of cytosolic prostaglandin E synthase in fetal-membrane and amniotic prostaglandin E2 accumulation in labor

PONE-D-20-30000R1

Dear Dr. Okuno,

We’re pleased to inform you that your manuscript has been judged scientifically suitable for publication and will be formally accepted for publication once it meets all outstanding technical requirements.

Kind regards,

Hai-Yan Lin

Academic Editor

PLOS ONE

Additional Editor Comments (optional):

Reviewers' comments:

Reviewer's Responses to Questions

**Comments to the Author**

1. If the authors have adequately addressed your comments raised in a previous round of review and you feel that this manuscript is now acceptable for publication, you may indicate that here to bypass the “Comments to the Author” section, enter your conflict of interest statement in the “Confidential to Editor” section, and submit your "Accept" recommendation.

Reviewer #1: All comments have been addressed

Reviewer #2: All comments have been addressed

2. Is the manuscript technically sound, and do the data support the conclusions?

Reviewer #1: (No Response)

Reviewer #2: Yes

3. Has the statistical analysis been performed appropriately and rigorously? 

Reviewer #1: (No Response)

Reviewer #2: Yes

4. Have the authors made all data underlying the findings in their manuscript fully available?

Reviewer #1: (No Response)

Reviewer #2: Yes

5. Is the manuscript presented in an intelligible fashion and written in standard English?

Reviewer #1: (No Response)

Reviewer #2: Yes

6. Review Comments to the Author

Reviewer #1: (No Response)

Reviewer #2: No additional comments. The authors have addressed the previous critiques. The value of the work is modest because of different sampling techniques between the two groups.

7. PLOS authors have the option to publish the peer review history of their article (what does this mean?). If published, this will include your full peer review and any attached files.

Reviewer #1: No

Reviewer #2: No

---

## [Editor Report · Acceptance letter]

15 Apr 2021

PONE-D-20-30000R1 

Up-regulation of cytosolic prostaglandin E synthase in fetal-membrane and amniotic prostaglandin E_2_ accumulation in labor 

Dear Dr. Okuno:

I'm pleased to inform you that your manuscript has been deemed suitable for publication in PLOS ONE. Congratulations! Your manuscript is now with our production department. 

Kind regards, 

on behalf of

Dr. Hai-Yan Lin 

Academic Editor

PLOS ONE